# Testosterone and Suicidal Behavior in Bipolar Disorder

**DOI:** 10.3390/ijerph20032502

**Published:** 2023-01-31

**Authors:** Leo Sher

**Affiliations:** 1James J. Peters VA Medical Center, Bronx, NY 10468, USA; leo.sher@mssm.edu; Tel.: +1-718-584-9000; 2Department of Psychiatry, Icahn School of Medicine at Mount Sinai, New York, NY 10029, USA; 3Department of Psychiatry, Columbia University Vagelos College of Physicians and Surgeons, New York, NY 10032, USA

**Keywords:** testosterone, suicide, bipolar disorder, mental health, public health

## Abstract

Bipolar disorder is associated with suicidal behavior. The risk of suicide for individuals with bipolar disorder is up to 20–30 times larger than that of the general population. Considerable evidence suggests that testosterone may play a role in the pathophysiology of suicidal behavior in both men and women with bipolar disorder and other psychiatric conditions. Testosterone has complex effects on psychological traits. It affects mood and behavior, including interactions with other people. Testosterone regulates pro-active and re-active aspects of aggression. Probably, both high and low levels of testosterone may contribute to the neurobiology of suicide in various patient populations. The effects of endogenous and exogenous testosterone on suicidality in patients with bipolar disorder need further investigation. The aim of this commentary article is to provide a commentary on the author’s work on the topic, summarize the literature on testosterone, bipolar disorder, and suicide, and encourage future research on this poorly studied topic.

## 1. Bipolar Disorder and Suicide

Bipolar disorder is a chronic psychiatric illness characterized by manic or hypomanic episodes alternating or combined with episodes of depression [1,2]. For bipolar I disorder, at least one manic episode must have taken place. Even though episodes of depression are usual, they are not needed for the diagnosis of bipolar disorder. At least one hypomanic episode and one major depressive episode are needed for the diagnosis of bipolar II disorder.

According to the National Comorbidity Survey replication in the U.S., the lifetime (and 12-month) prevalence estimates are 1.0% (0.6%) for bipolar I disorder, 1.1% (0.8%) for bipolar II disorder, and 2.4% (1.4%) for subthreshold bipolar disorder [3]. Around the world, a lifetime prevalence of bipolar disorder is over 1%, regardless of nationality, ethnicity, or socioeconomic status [1,4].

Underdiagnosis and misdiagnosis are frequent in individuals with bipolar disorder [5]. Patients with bipolar disorder are frequently diagnosed with depression, schizophrenia, or other psychiatric disorders. Differential diagnosis is sometimes difficult because the use/abuse of stimulants may result in manic symptoms.

Individuals with bipolar disorder are frequently functionally impaired [6]. Quality of life is significantly reduced in individuals with bipolar disorder, even when they are euthymic [6]. Cognitive deficits and physical comorbidities are associated with decreased quality of life in patients with bipolar illness.

Bipolar disorder is associated with suicidal behavior [1,7,8,9]. A review of suicide death in bipolar disorder indicates that the risk of suicide for individuals with bipolar disorder is up to 20–30 times larger than that of the general population [1,8]. It has been suggested that bipolar disorder is the psychiatric condition with the greatest frequency of suicidal behavior. The lifetime prevalence of suicide attempts is estimated to be 34% and 19% for women and men with bipolar disorder, respectively [10]. Suicide attempts are a significant predictor of suicide death [7].

Rapid cycling increases suicide risk in bipolar patients [1]. In addition, early onset bipolar disorder elevates suicide risk [11,12,13]. Some studies indicate that individuals in the U.S. have more genetic and familial risk factors for early onset bipolar disorder in comparison to people in Europe [13].

Our knowledge of the role of testosterone function and testosterone supplementation in the pathophysiology of suicidal behavior in individuals with bipolar disorder is limited. Studies of the role of testosterone in the pathophysiology of suicidality in bipolar disorder may potentially help to develop pharmacological interventions to reduce suicides in patients with bipolar disorder.

## 2. The Effect of Testosterone on Mood, Behavior and Suicidality

Testosterone is a steroid hormone that passes through cellular membranes and attaches to testosterone receptors prior to binding to DNA and enabling both RNA and protein synthesis [14,15,16]. Testosterone is the main androgen hormone controlling male sexual maturation and the maintenance of male secondary sexual features. In men, testosterone is mostly secreted by Leydig cells in testes, although some amount of testosterone is produced by the adrenal glands [14,15,16]. In women, lesser quantities of testosterone are secreted from the adrenals and ovaries.

Testosterone is transported to the brain via the blood circulation [17,18]. However, the enzymes necessary for testosterone biosynthesis are found in the brain, which suggests that steroid synthesis can occur in the brain. Testosterone acts via androgen receptors in the brain, or, after conversion by aromatase to estradiol, it acts via estrogen receptors in the brain. According to the current views, testosterone modulates the expression of certain genes by binding to androgen receptors in the brain. Further, acting via neurotransmitter receptors, testosterone has a non-genomic neuroactive effect [18,19]. Psychological effects of testosterone are related to the fact that some parts of the brain, including some areas of the hypothalamus and the amygdala, are affected by sex hormones. It has been suggested that social emotional behavior in men is modulated by testosterone via prefrontal-amygdala functional connectivity. It has also been suggested that an antidepressant effect of testosterone is related to its effects on the limbic system.

Testosterone has complex effects on psychological traits [20,21,22]. It affects mood and behavior, including interactions with other people. Testosterone regulates pro-active and re-active aspects of aggression. The administrations of testosterone to young females and to young males have an effect on behavioral and neural processes related to aggressive behavior [21]. It has been observed that aggression in high-dominance men and in men with low cortisol concentrations is affected by the administration of testosterone [21].

Success in competitive conditions, including situations encompassing aggression, can result in testosterone upsurge [21]. A string of successes can build up to substantially raise the testosterone levels of more competitive people.

Some studies indicate that high levels of testosterone are associated with increased rates of depression as well as hypomania [20]. Low levels of testosterone are linked to mood symptoms in some subgroups of depressed individuals.

In women of different ages, as in men, testosterone may influence mood and cognitive function including spatial perception, verbal learning, and memory [23,24]. For example, it has been shown that postmenopausal women taking transdermal testosterone gel and not taking other hormonal therapies significantly improved with regard to verbal learning and memory [24].

A significant number of studies suggest that testosterone is related to suicide risk [25,26,27,28,29,30,31,32,33]. Studies indicate that both high and low testosterone concentrations may play a role in suicidal behavior [31]. For example, we have observed that, among combat veterans, plasma free and total testosterone levels were lesser in suicide attempters in comparison to individuals without a history of suicide attempt [32]. In the same study, we have found that plasma free testosterone levels declined after dexamethasone administration amongst suicide non-attempters but not amongst suicide attempters, i.e., dexamethasone suppressed plasma free testosterone concentrations in non-attempters but not in attempters. Some studies, however, did not support a link between testosterone and suicidality [34,35]. For example, a study in Spain did not find a difference in testosterone levels between male suicide attempters and male healthy controls [34].

Methodological differences including dissimilarities in patient populations may contribute to differences in study results. Testosterone levels are measured by radioimmunoassays [32,36,37]. Intra-assay and inter-assay variabilities may affect the results of the studies. In addition, testosterone levels may be affected by the circadian pattern of circulating testosterone [36]. It has been reported that peak testosterone concentrations are observed at about 6 to 8 a.m. and nadir concentrations are seen at 6 to 8 p.m. However, the circadian pattern may be dissimilar in different individuals because the daily pattern of testosterone secretion is affected by age, stressors, and other factors [36,37].

About 3 years ago, researchers in Germany proposed the androgen model of suicide [38]. They suggested that in utero androgen exposure and adult androgen levels synergistically enable suicide. The authors described several lines of direct and indirect evidence, indicating that both a higher prenatal androgen load (with subsequent permanent neuroadaptations) and elevated adult androgen activity are involved in suicide death. The authors suggest that modifiable maternal behavioral traits such as smoking behavior, alcohol consumption, and stress levels contribute to the offspring’s prenatal androgen load and heighten the risk for suicide death later in life. The androgen model of suicide completion may help to develop new suicide preventive strategies in the future.

## 3. Testosterone and Suicidality in Bipolar Disorder

Our research indicates that testosterone may be involved in the neurobiology of suicidal behavior in individuals with bipolar disorder [28,29,33]. The impact of testosterone on suicidality may be related to the effect of testosterone on serotonin signaling and the hypothalamic–pituitary–adrenal (HPA) system [39]. Many research works have shown that the serotonergic system and the HPA axis play a role in the neurobiology of suicidal behavior [40]. It is important to note that cortisol modulates the serotonin system and neuronal survival.

We conducted the first study to investigate whether there is a connection between plasma testosterone levels and clinical characteristics in suicide attempters with bipolar disorder [28]. We restricted this investigation to bipolar suicide attempters to comprise a high-risk group for suicide attempt on follow-up and thus increasing statistical power. The number of major depressive episodes and the maximum lethality of suicide attempts were higher in men in comparison to women. Suicidal ideation scores were greater in women in comparison to men. Controlling for sex, we observed that plasma levels of testosterone positively correlated with the number of manic episodes and the number of suicide attempts.

We also conducted the first prospective study to assess whether plasma testosterone concentrations forecast suicide attempts in women with bipolar disorder and a history of suicide attempt [29]. Study participants were observed prospectively for up to 2.5 years. At baseline, plasma testosterone levels positively correlated with the number of previous major depressive episodes and suicide attempts. Greater baseline testosterone levels predicted suicide attempts during the follow-up period. An elevation in the plasma testosterone level by 0.1 ng/mL raised the probability of suicide attempt almost 17 times.

We also investigated whether greater plasma testosterone levels forecast future suicide attempts in men with bipolar disorder and a history of suicide attempt [33]. They were also followed up prospectively for up to 2.5 years. We found that greater baseline plasma testosterone levels predicted suicide attempts during the follow-up period. An elevation in the testosterone level by 1 ng/mL heightened the probability of suicide attempt almost two-fold.

Researchers in India examined blood testosterone and interleukin-17 concentrations and their relationship with suicidal behavior in patients with bipolar illness in remission [41]. The authors found no association of blood levels of testosterone or interleukin-17 with suicidal behavior. It is important to note that the sample of individuals with suicidal behavior was small.

## 4. Exogenous Testosterone Administration

Testosterone supplementation is widely used around the world [20,42]. In the U.S., testosterone preparation sales grew from around $100 million in 2000 to almost $2.7 billion in 2013 [41]. It has been reported that testosterone supplementation may trigger manic episodes in individuals with bipolar disorder [43]. Possibly, testosterone administration may lead to increase in self-violence and violence in individuals with bipolar illness. Some years ago, we reported a case of a middle-aged man with a history of bipolar disorder with psychotic features and polysubstance use disorder who had a violent episode several hours after getting a testosterone injection [44]. He punched his wife in the abdomen, which resulted in an internal bleeding and caused her death. It is not easy to determine a connection between the testosterone administration and suicidal or homicidal behavior. However, medical professionals should exercise caution when they prescribe testosterone supplementation to patients with bipolar illness. Patients with bipolar disorder who are prescribed testosterone need to be monitored for the emergence of symptoms of mania. They need to be assessed for mood lability, impulsive behavior, irritability, euphoria, insomnia, and regularly screened for suicidal and homicidal ideation or plan.

## 5. Conclusions

Available evidence suggests that testosterone may play a role in the pathophysiology of suicidal behavior in both men and women with bipolar disorder and other psychiatric conditions. Probably, both high and low levels of testosterone may contribute to the neurobiology of suicide in various patient populations [31]. Studies of the role of testosterone in the pathophysiology of suicidal behavior in bipolar disorder are important because endogenous and exogenous testosterone may affect suicidality in bipolar disorder [20,28,43,45]. The effects of endogenous and exogenous testosterone on suicidality in patients with bipolar disorder need further investigation. Future clinical neuroimaging studies including structural, functional, and pharmacological magnetic resonance imaging, and positron emission tomography may shed some light on the role of testosterone in the biological mechanisms of mood disorders and suicidal behavior. About 60% of all individuals dying from suicide have mood disorders [46]. Therefore, studies of the neurobiology of suicidal behavior in persons with bipolar illness and other mood disorders are of the utmost importance.

At the present time, our knowledge of the pathophysiology of suicide remains fragmented. There are no proven biological or clinical biomarkers that can be employed for the evaluation of suicide risk. It is important to note that suicide risk may extend beyond psychiatric disorders. Physicians, scientists, and policy-makers should try to develop new programs to better understand the phenomenon of suicide including the neurobiology of suicide. Studies of the psychological and biological mechanisms of suicidality may enable physicians and researchers to design more effective suicide prevention programs.

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
