# Peer review of "Testosterone and Suicidal Behavior in Bipolar Disorder"

_ijerph, 2023, doi:10.3390/ijerph20032502_

Round 1
Reviewer 1 Report
This manuscript, entitled "Testosterone and suicidal Behavior in bipolar Disorder" consists of a brief narrative review about the possible association between plasma testosterone levels, testosterone administration and suicidality in people with bipolar disorder. It is well written, and the author has done a lot of work on the subject, with contributions going back decades.
The topic of the article (suicide in bipolar disorder) is of great interest to the journal, as it is a serious public health problem, in which biological factors can be critical. The neuro-psycho-endocrinological approach can shed light on the complexity of the suicidal phenomenon.
However, this association is very controversial. The manuscript emphasizes the studies (mainly by the author of the commentary) that point to this association. Studies are certainly also mentioned (Butterfield et al., 2005; Perez-Rodriguez et al., 2011) in the opposite direction, but the overall view is unbalanced. Being a narrative review, there is a risk of biasing the selection of articles in favor of the proposed hypothesis.
The reader may be confused by the conflicting results of the studies. Some show that “among combat veterans, plasma free and total testosterone levels were lower in those who attempted suicide (Sher et al., 2021).” But a significant number of studies suggest that testosterone is related to suicide risk (Roland et al., 1986; Tripodianakis et al., 2007; Markianos et al., 2009; Sher, et al., 2012; 81 Sher et al., 2014; Zhang et al., 2015; Sher, 2018; Sher et al., 2021; Sher et al., 2022). I would recommend a clarification of the "state of the art", perhaps separating suicide attempts from completed suicide, or investigating the methodological issues that sustain these contradictions.
He would recommend including a paragraph on the difficulties of measuring plasma testosterone levels throughout the day, and the different ways to do it.
I would recommend expanding the information about the influence of testosterone and mood in women, throughout their life cycle. Women have twice the prevalence of depression than men, and a higher incidence of suicide attempts, despite having clearly lower testosterone levels.
Evidence about the effect of synthetic testosterone on suicidality in patients with BD is very limited, despite being the focus of the article .
In the abstract I would avoid talking about “our studies”, referring to those published by the author of the review. In fact, given the controversy on the subject, I would avoid including the phrase "Our prospective studies have shown that higher baseline plasma testosterone levels predict suicide attempts during the follow-up period in patients with bipolar disorder."
The content of the abstract is focused on the effects of testosterone on behavior and mood. I would recommend focusing more on the purpose of the review, emphasizing the scant evidence in this regard.
In summary, the main virtue of this narrative review on a controversial association is to encourage future research on the matter.
Reviewer 2 Report
This is an interesting brief commentary of the author's work to understand the role of testosterone in the experience of bipolar disorder, and the experience of suicidality amongst people living with bipolar disorder.
I have a number of suggestions to improve the quality of the manuscript, and to improve the framing of the article as a commentary piece for a broader audience.
Abstract - it would be useful to have a sentence in the abstract outlining the aims of the commentary article (i.e. to provide a commentary of the author's work on the topic and summarise the literature on testosterone, bipolar disorder, and suicide). This would also help position the commentary as a more selective (i.e., non-systematic) review of the literature.
There also needs to be a earlier statement on why understanding the relationships/interplay between testosterone, bipolar disorder, and suicide, is needed and important (at present, this comes a bit late into the article which doesn't aid the readability of the author's work/ideas). This could be a brief paragraph at the start of the commentary or at the end of the first section on Bipolar Disorder and suicide (e.g. along the lines of 'this is what we know about BD and suicide... studying testosterone's role in this relationship would be useful because...' and then go into more detail). The present manuscript feels like it needs a bit more framing around why this topic is of importance and make parts of the manuscript feel a bit less 'bitty'/disconnected, especially for readers who are not intimately familiar with this research area.
Line 58 - a minor point but I would suggest changing "go across" to "passes through" ('go across' seems a bit informal/non-specific).
Line 77 - there is mention of testosterone being associated with an increased risk for depression and hypomania but it is unclear why this would be the case. Some further details on the mechanisms/pathways involved would be useful (for a general reader who might not be an expert on these mechanisms).
In relation to the above, it would be useful to have additional detail on the mechanisms/pathways implicated in the testosterone-suicidality/mood link throughout the commentary - it sometimes feels like understanding of these links are assumed on behalf of the reader.
Line 94 - I was interested in this androgen model of suicide the author mentions, it would be useful to have some additional details on what this model suggests and why in utero/adult exposure would lead to increased suicidality/suicide risk (esp. in utero exposure).
Whilst I appreciated the author's discussion of their own work in this area, it may be useful to have some reflection in this commentary piece on the challenges associated with this area of work (Line 97 onwards). I would also appreciate a clearer introduction to this section on the author's own work in terms of setting up their own lab's studies (e.g. "Our research has shown...) - at present, this section seems to start very abruptly without a clearer introduction for the benefit of the reader.
Line 121 - the author mentions some interesting work by Indian researchers but I was unsure if this is collaborative work between the commentary author and these other researchers, or a separate lab's work. Please clarify / amend as appropriate.
Line 131 - it would be useful here to explain why testosterone would lead to increased self-violence (picking up my point earlier about the mechanism of action)
It would be of interest to have a clearer section on Clinical Implications (which the author briefly mentions around Line 138) - given the issues with testosterone administration and testosterone's effects on mood/behaviour, a discussion of the challenges for clinicians would be very welcome here and would be of interest to a wide readership (and would help to build to the conclusion of the commentary).
I would also appreciate a commentary from the author towards the end of the article about the future directions for this literature.
Finally, a general thought for the author is to ensure that the commentary is written to be more readable to a broader audience - at present, the work could 'flow' better and provide a little more detail in places regarding how testosterone works and why some of the above points are important (e.g. why high/low testosterone seem important for outcomes, how testosterone works in terms of mood/cognition/behaviour) and a clearer statement in the Conclusion regarding why it is important and clinically useful to consider testosterone's role in suicidality amongst people living with bipolar disorder.
Round 2
Reviewer 2 Report
Thank you to the author for addressing my comments in a clear and systematic manner. I have no further comments and feel that this is a nice commentary of the author's work in this area that will be of interest to the journal's readership.